

# The importance of open science for biological assessment of aquatic environments

Marcus W. Beck[1,2], Casey O'Hara[3], Julia S. Stewart Lowndes[4], Raphael D. Mazor[1], Susanna Theroux[1], David J. Gillett[1], Belize Lane[5] and Gregory Gearheart[6]

[1] Southern California Coastal Water Research Project, Costa Mesa, CA, USA
[2] Tampa Bay Estuary Program, Saint Petersburg, FL, USA
[3] Bren School of Environmental Sciences & Management, University of California, Santa Barbara, CA, USA
[4] National Center for Ecological Analysis and Synthesis, Santa Barbara, CA, USA
[5] Department of Civil and Environmental Engineering, Utah Water Research Laboratory, Utah State University, Logan, UT, USA
[6] California Water Resources Control Board, Sacramento, CA, USA

## ABSTRACT

Open science principles that seek to improve science can effectively bridge the gap between researchers and environmental managers. However, widespread adoption has yet to gain traction for the development and application of bioassessment products. At the core of this philosophy is the concept that research should be reproducible and transparent, in addition to having long-term value through effective data preservation and sharing. In this article, we review core open science concepts that have recently been adopted in the ecological sciences and emphasize how adoption can benefit the field of bioassessment for both prescriptive condition assessments and proactive applications that inform environmental management. An example from the state of California demonstrates effective adoption of open science principles through data stewardship, reproducible research, and engagement of stakeholders with multimedia applications. We also discuss technical, sociocultural, and institutional challenges for adopting open science, including practical approaches for overcoming these hurdles in bioassessment applications.

## INTRODUCTION

Bioassessment is an essential element of aquatic monitoring programs that helps guide decisions for managing the ecological integrity of environmental resources. Legal mandates to assess biological condition have stimulated the development of bioassessment programs and tools in the United States (Clean Water Act, CWA), Canada (Canada Waters Act), Europe (Water Framework Directive), China (Environmental Quality Standards for Surface Water), South Africa (National Water Act), and elsewhere (*Borja et al., 2008*). Decades of research have supported the development of assessment indices for multiple assemblages with regional applications in streams, rivers, lakes, and marine environments (*Karr et al., 1986*; *Kerans & Karr, 1994*; *Fore & Grafe, 2002*; *Beck & Hatch, 2009*; *Borja, Ranasinghe*

Corresponding author
Marcus W. Beck, mbeck@tbep.org

& Weisberg, 2009; Borja et al., 2016). Substantial technical advances have been made in measuring biological responses to environmental change (Hawkins et al., 2000a; Hawkins et al., 2000b), how these responses can be distinguished from natural environmental variation (Stoddard et al., 2006; Hawkins, Olson & Hill, 2010), and interpreting the impacts of these changes (Davies & Jackson, 2006).

Integrating bioassessment products (e.g., scoring indices, causal assessment protocols) into management or regulatory frameworks can be challenging, despite the technological advances (Kuehne et al., 2017). How a bioassessment product is used in practice to inform decisions and prioritize management actions can differ from why it may have been originally developed. Numerous assessment products have been developed for specific regional applications (Birk et al., 2012) and concerns about redundancy, comparability, duplicated effort, and lack of coordinated monitoring have recently been highlighted (Cao & Hawkins, 2011; Poikane et al., 2014; Kelly et al., 2016; Nichols et al., 2016). Kuehne, Strecker & Olden (2019) recently highlighted a lack of institutional connectivity among actors with expertise in freshwater assessment as a hallmark of the status quo in which applied science is conducted. Moreover, existing indices may not be easily calculated by others beyond initial research applications (Hering et al., 2010; Nichols et al., 2016) or may be incorrectly applied based on differences between goals for developing an index and the needs of management programs (Dale & Beyeler, 2001; Stein et al., 2009). The abundance of available products can be a point of frustration for managers given a lack of guidance for choosing among alternatives, particularly as to how different assessment products relate to specific management, monitoring, or policy objectives (Dale & Beyeler, 2001; Stein et al., 2009).

To address these challenges, a new mode of operation is needed where method development is open and transparent, developed products are discoverable and reproducible, and most importantly, implementation in the management community is intuitive and purposeful. Open science principles that improve all aspects of the scientific method can help meet these needs and there is a unique opportunity in bioassessment to leverage openness to support public resources. Open science and its ideals originated partly due to failures of reproducibility and biases in the primary literature that were revealed as systematic concerns in research fields with immediate implications for human health (Makel, Plucker & Hegarty, 2012; Franco, Malhotra & Simonovits, 2014). These ideas and the failures that they address have slowly permeated the ecological and environmental sciences (Hampton et al., 2015; Hampton et al., 2016; Lowndes et al., 2017). Open science has also influenced how research workflows are conceptualized in other disciplines (e.g., archaeology, Marwick et al., 2016, behavioral ecology, Ihle et al., 2017, hydrology, Slater et al., 2019, vegetation sciences, Collins, 2016) and has enabled a shift towards publishing structures that are more fair and transparent through open access (Van Oudenhoven, Schröter & De Groot, 2016; Essl et al., 2020). Limited examples have suggested that open access databases can be leveraged to develop bioassessment products that increase transparency among stakeholders (Borja et al., 2019). Adopting an open science paradigm in bioassessment is particularly relevant compared with other fields given the explicit need to develop products that are accessible to the management community.

Legal and ethical precedents in bioassessment may also necessitate open data sharing given that environmental monitoring programs are often established to protect and maintain publicly-owned natural resources.

## SURVEY METHODOLOGY AND OBJECTIVES

This review draws on previous literature to describe approaches for open science that can empower the research and management community to embrace a new mode of thinking for bioassessment applications. These approaches are expected to benefit the bioassessment research community by providing new tools that augment existing workflows for developing assessment products and improving their ability to address environmental issues by bridging the gap between the scientific, management, and regulatory communities. The intended audience for this review is primarily the research team that develops bioassessment products, but we also write for the funders and users (e.g., regulators and managers) of these products to emphasize the value of investing in open science for the protection of public resources.

This traditional review covers literature published in recent years advocating for open science in different fields of study. Because no similar efforts have yet been made to apply these principles to bioassessment, we draw on examples from the previous literature that demonstrate successful applications in other fields to motivate researchers and practitioners to embrace these new ideas in bioassessment. Comprehensive and unbiased coverage of the previous literature was accomplished by querying online search engines, primarily Google Scholar, with search terms as they relate to open science (e.g., ''reproducibility'', ''data science'', ''open source'') and with Boolean operators to find applications to bioassessment (e.g., ''reproducibility AND bioassessment''). Studies were included if they provided general overview of open science concepts that were relevant to bioassessment or if they directly described open science applications to bioassessment, although the latter were scarce. Emphasis was given to the breadth of research that has supported the development of open source software applications that can aid bioassessment, both as general tools and more specific programs tailored for indicator development. We excluded studies that described applications with citizen science components. Although citizen science can be a valuable tool for researchers and managers, methods for effective implementation are beyond the scope of this review.

Our objectives are to (1) provide a general overview of principles of open science and (2) empower the research community by providing examples of how these principles can be applied to bioassessment. For the second objective, we also provide a case study of stream bioassessment in the urban landscape of southern California to demonstrate a successful proof of concept. Herein, open science 'tools' describe best practices and specific applications that use an open philosophy to support applied science. We structure the review by first introducing open science principles, then describing how these principles could be applied to bioassessment (i.e., developing goals, curating data, and applying open-source software) including a case study example, and lastly providing a discussion of limitations and opportunities to better contextualize real world applications of open science.

**Table 1** **Core definitions and principles of open science.** Content adapted from Open Knowledge International, http://opendefinition.org/, Creative Commons, https://creativecommons.org/about/program-areas/open-science/, D. Gezelter, http://openscience.org/, and *Powers & Hampton (2019)*.

| Concepts and principles | Description |
| --- | --- |
| Open | Anyone can freely access, use, modify, and share for any purpose |
| Open Science | Practicing science in such a way that others can collaborate and contribute, where research data, lab notes and other research processes are freely available, under terms that enable reuse, redistribution and reproduction of the research and its underlying data and methods |
| Reproducible | Producing equivalent outcomes from the same data set, or in the case of computational reproducibility, producing equivalent outcomes from the same data set using the same code and software as the original study |
| Principle 1 | Transparency in experimental methods, observations, and collection of data |
| Principle 2 | Public availability and reusability of scientific data |
| Principle 3 | Public accessibility and transparency of scientific communication |
| Principle 4 | The use of web-based tools to facilitate scientific collaboration and reproducibility |

## Principles of open science

Conventional modes of creating scientific products and more contemporary approaches that align with open science principles share the same goals. Both are motivated by principles of the scientific method that make the process of discovery transparent and repeatable. Where the conventional and open science approaches diverge is the extent to which technological advances facilitate the entire research process. Distinction between the two approaches can be conceptualized as the "research paper as the only and final product" for the conventional approach, whereas the open science approach is inherently linked to advances in communication and analysis that have been facilitated by the Internet and computer sciences (Table 1). As a result, the open science approach can enhance all aspects of the scientific process from initial conception of a research idea to the delivery and longevity of a research product (Fig. 1). The process is iterative where products are improved by the individual and/or others, facilitated by open science tools that enhance access and reproducibility of data.

The paradigm of the research paper as a final scientific product can inhibit the uptake of research methods and findings by environmental managers. The research paper is conventionally viewed as a communication tool for scientists to report and share results among peers. Researchers access periodicals to stay informed of scientific advances and use the information to replicate and improve on methods for follow-up analysis. Although the primary literature continues to serve this critical role, this workflow is problematic when scientific products are needed to serve interests outside of the research community. For example, the paper as an endpoint for environmental managers fails to deliver products that are easily accessible from the practitioner's perspective, both in application and
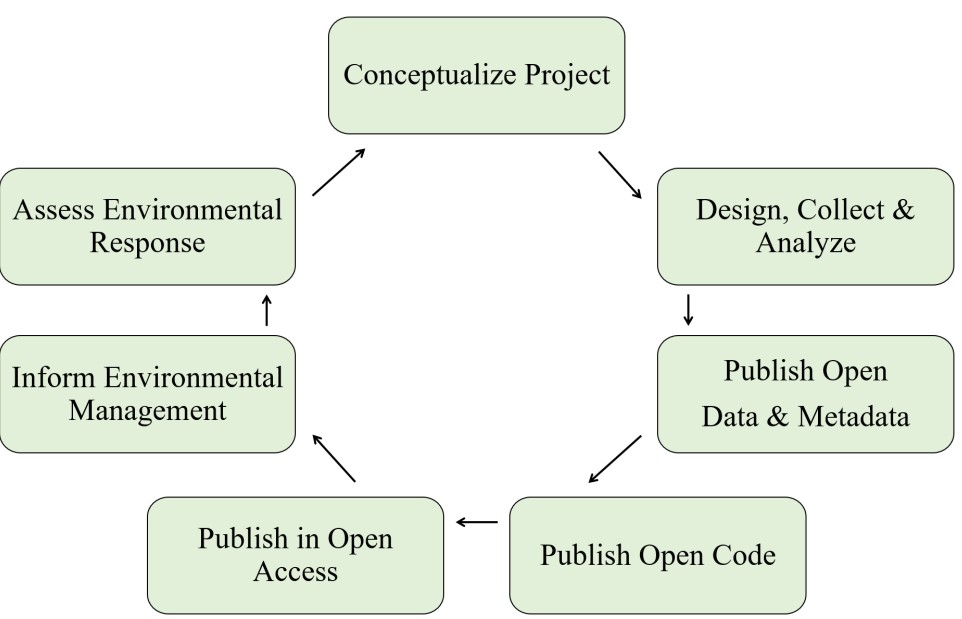

**Figure 1** **A simplified workflow of the open science paradigm (adapted from *Hampton et al., 2015*).** All aspects of the research process, from the conception of an idea to publishing a product, can be enhanced using open science tools. The workflow is iterative where products are continually improved through collaborations facilitated through discovery and reproducibility of open data.

interpretation. A research paper is less likely to effect environmental change because it does not provide a mechanism to transfer actionable information to those that require scientific guidance for decision-making, such as sharing analysis code or results that describe output from assessment products. Numerous studies have documented implementation failures as a result of siloing among research communities where the flow of information does not extend beyond institutional walls (e.g., *Mitchell, 2005*; *Liu et al., 2008*). Information loss over time is another concern associated with the paradigm of research paper as final product (*Michener et al., 1997*), particularly as intimate knowledge of study details is lost as new projects are initiated or individuals leave institutions.

## Open data as a component of the open science process

Open data is a fundamental component of the broader open science process described in Fig. 1. Under this mode of thinking, the research team becomes stewards of its data. For bioassessment data, government institutions may be the primary stewards of information that supports product development within a broader research team. Stewardship allows the data to be treated as a dynamic product with a traceable and replicable provenance (i.e., origin), rather than proprietary and serving only the internal needs of an immediate research goal. Metadata that describe the structure and history of a dataset ensure the data have an identity. Metadata also encourage adoption of core data structures that allow integration across different sources, which is critical for collaboration across institutional boundaries (*Horsburgh et al., 2016*; *Hsu et al., 2017*). Other open science practices, such as integration of data with dynamic reporting tools or submitting data to a federated repository (i.e., a
decentralized database system for coordination and sharing), can facilitate communication for researchers and those for which the research was developed (*Bond-Lamberty, Smith & Bailey, 2016*). Prominent examples that can benefit next-generation bioassessment methods, such as molecular-based techniques for species identification, include the BarCode of Life Data Systems (BOLD) and GenBank repositories.

Open data can benefit research by contributing to an increase in novel products created through collaboration. Collaborative publications have increased in the environmental sciences as research teams leverage open data to create synthesis products that allow novel insights from comparisons across multiple datasets. Quantitative meta-analyses and systematic reviews are increasingly used to extract information from the primary literature (*Carpenter et al., 2009*; *Lortie, 2014*). In addition, open data products can increase efficiency of the individual researcher and a collective research team by encouraging collaborators to adopt an open science workflow. Many tools developed within the software and computer science community to facilitate open process and the creation of open data are now easily accessible to environmental scientists (*Yenni et al., 2019*). Version control software (e.g., Git, GitHub), open source programming languages (e.g., R, Python), and integrated development environments (IDEs, e.g., RStudio, Spyder) can all be leveraged to dynamically create and share open data products that can build institutional memory. These tools promote deliberate and shared workflows among researchers that can lead to better science in less time (*Lowndes et al., 2017*) and have proven useful in recent applications in the hydrologic sciences (*Idaszak et al., 2017*; *Slater et al., 2019*).

Open access to data can also benefit management and regulatory communities. Openness can improve the value of data from monitoring programs by facilitating data discovery and synthesis, often through the adoption of a common metadata structure and integration of data within federated data networks (e.g., DataONE, iRODS). Research institutions can also use open data maintained by management or regulatory communities to develop products that directly support the mission of the latter, e.g., assessment methods developed from long-term monitoring datasets that identify priority areas to focus management actions or fulfill regulatory obligations. Open data can also improve public trust in scientific findings by exposing the underlying information used to develop a research product (*Grand et al., 2012*). Similar concepts are used in "blockchain" technologies that allow public financial transactions in an open, distributed format, as for trading in cryptocurrencies (*Pilkington, 2016*). Increased trust could facilitate eventual adoption of proposed rules or regulations that are based on research products created from open data. More efficient and effective implementation of potential regulations may also be possible if supporting data are openly available.

## Applying open science principles to bioassessment

Here we provide a detailed description of open science processes that the bioassessment community could leverage to create reproducible, transparent, and discoverable research products for environmental managers. The below examples require understanding the distinction between the general open science process in Fig. 1, open data as an individual component of the open science process, and the technology-based tools that can be used

to achieve these ends. Both the tools and open data are critical components that facilitate the broader process to achieve the principles outlined in Table 1. "Openness" of process, tools, and data exists on a continuum, and incremental improvements can transform an individual's and research group's practice over time. We encourage awareness that an open process adopts the open science tools that are appropriate for a research question and the creation of open data can be a fundamental component of the process. Acceptance by the research team and collaborators of the concepts described in Table 1 is critical to achieving openness.

The overall process is shown in Fig. 2 as an expansion of general concepts in Fig. 1. This iterative flow of information is facilitated by (1) openly sharing planning documents, (2) using established metadata standards to document synthesized data products, (3) hosting data products on open repositories, (4) creating reproducible summary documents that integrate the data and research products, and (5) incorporating the developed product into interactive applications that deliver the results to the managers and stakeholders. The technical phase of defining research goals, collecting and synthesizing data, and developing the bioassessment product are primary tasks of the research team. However, the open science process is distinguished by the flow of information to and from the research phase that can benefit the specific project and the science of bioassessment as a whole.

## Developing bioassessment goals

In an open science process, the goals identified by the research team for developing a bioassessment product should occur through direct, two-way interaction with the management or regulatory institution that requires the product. Although such an approach has historically been used to develop bioassessment products, the interaction in an open science workflow differs in how information is exchanged. This exchange can be accomplished through direct communication and sharing of planning documents to ensure all decisions are transparent, i.e., open planning. In person meetings are ideal, but planning documents are dynamic and will require remote sharing and revision as ideas progress. Online tools such as Google documents, Slack discussion channels, and open lab notebooks can be instrumental for collaboration. More informal approaches, such as blogging and sharing ideas on social media, can expose new concepts to the broader community for guidance (*Woelfle, Olliaro & Todd, 2011*; *Darling et al., 2013*). Overall, the research team should use these tools to identify stakeholder needs while also considering the balance between the research goals and limitations of the data to meet these goals. This approach will ensure that the needs of the management and stakeholder communities will be consistent with the services provided by the research product.

An important practice that is not often used in bioassessment for project planning is study pre-registration. This is a relatively new addition to the philosophy of open science that allows a research team to define their study procedures, expected outcomes, and statistical analysis plans in advance of the actual study (*Munafó et al., 2017*). Although the standard scientific method may seem to support such proactive practices, pre-registration is an explicit declaration to make the intentions of a study design clear to avoid publication bias where only positive outcomes are reported and to prevent an interpretive result where
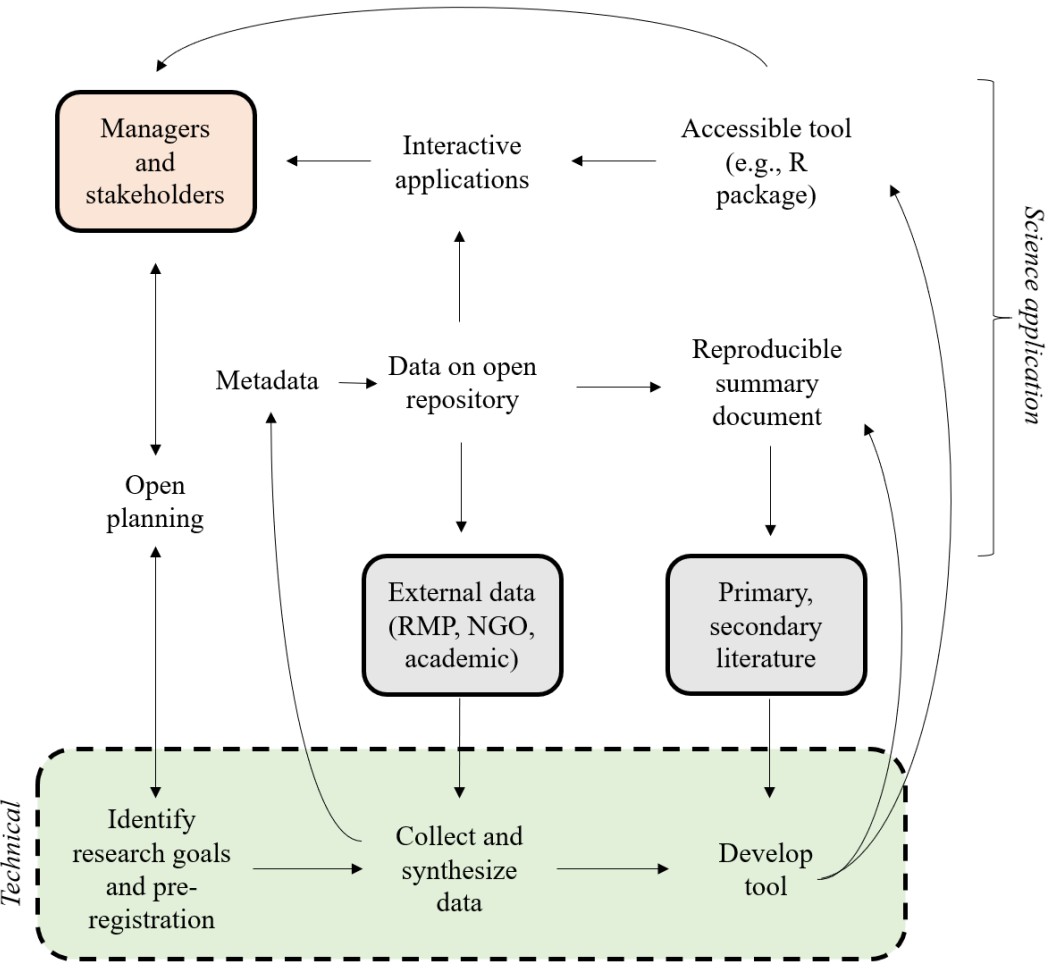

**Figure 2   An idealized open approach for bioassessment applications.** The green box represents the technical steps of the individual research team for developing the product, the manager and stakeholder box are those that require or motivate the creation of bioassessment products, the gray boxes indicate sources of external information (data and guidance documents) as input into the technical process, and the open text indicates open components of the planning, application, or implementation phase of a bioassessment product. The figure was adapted from *Hampton et al. (2015)*. NGO: non-government organization, RMP: regional monitoring program.

the researcher retrospectively defines study objectives after initial results are obtained if they do not agree with expectations. This latter issue is a serious concern where scientists use postdiction with significant hindsight bias in place of prediction and conventional hypothesis testing to inform scientific discovery (*Nosek et al., 2018*). Registered reports can also be used as a publishing format where an initial study design is peer-reviewed and the article is provisionally accepted by a journal if the results are created with methods that do not deviate from the accepted design. This promotes greater coverage in the primary literature of null conclusions that otherwise may not have been published, reducing bias for publishing positive results.

Pre-registration has been used extensively in clinical research (*Dickersin & Rennie, 2003*), where outcomes often have immediate implications for human health and well-being. In contrast, bioassessment studies often focus on developing applied products, where conventional hypothesis testing is less a concern. Studies are typically methods-focused where a research product is developed to address a management or regulatory need, rather than a specific research question with a testable hypothesis. However, pre-registration could be an important tool for the environmental sciences where an explicit declaration of study intent as being applied or methods-focused could prevent postdiction or an otherwise misuse of study results after a project is completed. Existing venues that support pre-registration of studies across multiple disciplines could be used in bioassessment study planning (e.g., Open Science Framework, AsPredicted).

## Curating bioassessment data

After project goals are established, the research team identifies requirements and sources of data that need to be synthesized to meet the research needs. Bioassessment data, or more generally, biological data obtained from field sampling have a unique set of challenges that require added vigilance in data stewardship (*Cao & Hawkins, 2011*). Species identification requires a tradeoff between taxonomic specificity and cost (*Lenat & Resh, 2001*; *Chessman, Williams & Besley, 2007*). Species names also change regularly requiring updates to standard taxonomic effort (STE) tables that are critical for many biological indices, although some standardized databases have facilitated broad-scale comparisons (e.g., World Register of Marine Species, *Costello et al., 2013*). Unidentified or ambiguous taxa must also be explicitly treated in analysis workflows (*Cuffney, Bilger & Haigler, 2007*), e.g., are they treated as missing values or are they substituted with coarser taxonomic designations? In contrast, molecular techniques based on DNA barcoding eliminate the need for direct species collection and morphological identification (*Deiner et al., 2017*; *Hering et al., 2018*). These next-generation approaches have capitalized on advances in database development that allow open access by diverse researchers across disciplines and are well-suited for the development of additional open science tools. Despite these advances, molecular-based approaches have also suffered from challenges related to standardization of workflows and coverage of reference databases (*White et al., 2014*; *Elbrecht et al., 2017*).

Open science tools can facilitate the curation of bioassessment data by addressing the above challenges. For example, a multimetric index may require taxonomic data collected at multiple sites by different institutions, whereas the output data may include summary scores, individual metrics, and any additional supporting information to assess the quality of the output. In an open science workflow, these data products can be documented using a standardized metadata language (e.g., Ecological Metadata Language Standard, or EML) which describes the who, what, and why to ensure the data have an identity. Adoption of a metadata standard also ensures that a machine-readable file is produced to allow integration into a data repository. This will allow a synthesized data product to be discoverable beyond the specific research application and will provide metadata to help others understand the context of the data (e.g., *Idaszak et al., 2017*). Finally, the dataset can be assigned a unique digital object identifier (DOI, e.g., through Zenodo) that provides a

permanent address and is also citable to allow researchers to track usage of a bioassessment data product.

In an open paradigm, the data itself is a product to achieve the research goals and also becomes available to the research and management community as a fully documented source of information that has value beyond the specific project. The openness of the synthesized data product is one of the primary means of facilitating the application of a bioassessment product. The synthesized data product can be used by the research team to create interactive applications for stakeholders to share and explore the data and is also fully integrated into summary reports using software for generating dynamic documents (e.g., using knitr, *Xie, 2015*, RMarkdown, *Allaire et al., 2018*, Jupyter notebooks, *Kluyver et al., 2016*). Continuous integration services can automate quality control and regularly update data products as new information is collected (*Yenni et al., 2019*). The data product also becomes available on an open data repository that is discoverable by other researchers and can contribute to alternative scientific advances beyond the immediate goals (e.g., Hydroshare for the hydrologic sciences, *Idaszak et al., 2017*).

## Using R for bioassessment application

The R statistical programming language (*RDCT, 2020*) is one of the most commonly used analysis platforms in the environmental sciences (*Lai et al., 2019*; *Slater et al., 2019*) and many existing R packages have value for the bioassessment community (Table 2). For managing the day to day tasks of working with multiple datasets, the tidyverse suite of packages provides the necessary tools to import, wrangle, explore, and plot almost any data type (*Wickham, 2017*). The tidyverse also includes the powerful ggplot2 package that is based on a syntactical grammar of graphics for plotting (*Wilkinson, 2005*; *Wickham, 2009*). This package provides a set of independent plotting instructions that can be built piecewise and is a departure from other graphics packages that represent a collection of special cases that limit the freedom of the analyst. In bioassessment, ggplot2 can be used both in an exploratory role during the development phase and also to create publication quality graphics.

Bioassessment data are inherently spatial and recent package development has greatly improved the ability to analyze and map geospatial data in R. The raster package can used to read/write, manipulate, analyze, and model grid-based spatial data (*Hijmans, 2019*), which are often common supporting layers for bioassessment (e.g., elevation or climate data). For vector data (i.e., points, lines, and polygons), the sfpackage ("simple features", *Pebesma, 2018*) uses principles of data storage that parallel those from the tidyverse by representing spatial objects in a tidy and tabular format. This facilitates analysis by presenting complex spatial structures in a readable format that can be integrated in workflows with existing packages, including other mapping packages (e.g., leaflet, *Cheng, Karambelkar & Xie, 2018*, or mapview, (*Appelhans et al., 2018*)). This allows the research team to use a workflow that is focused in a single environment, rather than using separate software for statistical and geospatial analysis.

R is fundamentally a statistical language and several existing R packages can be used to evaluate and support bioassessment data. Random forest models have been used

**Table 2  R packages that can be used in the development and application of bioassessment products.**

| Task | Package | Description |
|---|---|---|
| General | tidyverse (*Wickham, 2017*) | A suite of packages to import, wrangle, explore, and plot data. Includes the popular ggplot2 and dplyr packages. |
| Mapping, geospatial | sf (*Pebesma, 2018*) | A simple features architecture for working with vectorized spatial data, including common geospatial analysis functions |
| | raster (*Hijmans, 2019*) | Reading, writing, manipulating, analyzing, and modeling gridded spatial data |
| | leaflet (*Cheng, Karambelkar & Xie, 2018*) | Integration of R with the popular JavaScript leaflet library for interactive maps |
| | mapview (*Appelhans et al., 2018*) | Creates interactive maps to quickly examine and visually investigate spatial data, built off leaflet and integrated with sf |
| Statistical modeling | randomForest (*Liaw & Wiener, 2002*) | Create classification and regression trees for predictive modeling |
| | nlme (*Pinheiro et al., 2018*) | Non-linear, mixed effects modeling |
| | mgcv (*Wood, 2017*) | Generalized additive modeling |
| Community analysis | TITAN2 (*Baker, King & Kahle, 2015*) | Ecological community threshold analysis using indicator species scores |
| | indicspecies (*De Caceres & Legendre, 2009*) | Indicator species analysis |
| | vegan (*Oksanen et al., 2018*) | Multivariate analysis for community ecology |
| Science communication | shiny (*Chang et al., 2018*) | Reactive programming tools to create interactive and customizable web applications |
| | rmarkdown (*Allaire et al., 2018*) | Tools for working with markdown markup languages in .Rmd files |
| | knitr (*Xie, 2015*) | Automated tools for markdown files that process integrated R code chunks |

to develop predictive bioassessment indices that compare observed taxa to modeled expectations (i.e., O/E indices). The randomForest package (*Liaw & Wiener, 2002*) uses an ensemble learning approach that is robust to complex, non-linear relationships and interactions between variables. These models are particularly useful with large, regional datasets that describe natural and anthropogenic gradients in condition (*Laan & Hawkins, 2014*; *Mazor et al., 2016*). The nlme package can be used to create non-linear mixed effect models that are more flexible than standard regression approaches (*Pinheiro et al., 2018*). The nlme package can develop models for nested sampling designs, such as repeat visits to sample sites or otherwise confounding variables that contribute information but are not unique observations (*Mazor et al., 2014*). The mgcv package provides similar functionality as nlme, but uses an additive modeling approach where individual effects can be evaluated as the sum of smoothed terms (*Wood, 2017*). The mgcv package is often applied to model biological response to stressor gradients (*Yuan, 2004*; *Taylor et al., 2014*).

Other R packages have been developed specifically for bioassessment. For example, the TITAN2 package can be used to develop quantitative evidence of taxon-specific changes in abundance and occurrence across environmental gradients (*Baker, King & Kahle, 2015*). Results from this package can support exploratory analysis for developing bioassessment

products, such as identifying indicator species or establishing numeric criteria (*Taylor et al., 2018*). The results can be also be used post hoc to evaluate potential response of a biological index with changing environmental conditions, such as proposed management actions for rehabilitation (*King et al., 2011*). Alternatively, the `indicspecies` package provides similar functionality but is based only on species occurrence or abundance matrices across sites (*De Caceres & Legendre, 2009*). This package can be used to identify species at sites if continuous environmental data are unavailable, such as those that are representative of reference conditions (*Bried et al., 2014*). Finally, the `vegan` package has been a staple among community ecologists for multivariate analyses in R (*Oksanen et al., 2018*).

Although the R network includes over 15,000 user contributed packages, only a handful of these packages are specific to bioassessment. Community practices have allowed R to reach new audiences where new packages build on the work of others and are transportable between users and operating systems. Formalized communities, such as rOpenSci, encourage standardization and review of contributed packages within the ecological sciences to make scientific data retrieval reproducible. Several tools have also been developed and published in the last five years that greatly simplify the process of creating new packages in R (*Wickham, 2015*; *Wickham, Hester & Chang, 2018*). The advantages of creating and sharing R packages that are specific to bioassessment applications are important for several reasons. First, an R package compartmentalizes technical instructions developed during the research phase that can be executed by anyone with access to the software. R packages also require explicit documentation of the functions and data requirements. As such, package users will not only have access to underlying code but also understand the why and what for different package functions.

Finally, R can be used to create interactive applications that deliver bioassessment products to stakeholders and managers in entirely novel contexts. In particular, the `shiny` package provides programming tools built around concepts of reactivity, where data inputs and outputs can be modified in real time (*Chang et al., 2018*). A `shiny` application is an interactive user interface that is developed with R code, but is a standalone product that can be used without any programming experience. These applications are deployed online and can extend the reach of bioassessment products to those that require the information for decision-making but otherwise do not have the time or resources to learn R. Applications built in `shiny` can also be easily linked to other R packages. For example, a `shiny` website could be created to allow users to upload raw data and estimate and report bioassessment scores using an R package developed externally. Moreover, `shiny` applications are completely customizable and can be tailored by the developer to the specific needs of any user. This distinction separates `shiny` from other web-based analysis platforms.

## Open science in practice: the SCAPE project

Although bioassessment products have been sufficiently developed in California (USA), there are no narrative or numeric criteria in place to support designated aquatic life uses in wadeable streams, nor are bioassessment data actively used to support conservation or watershed management. Indices using benthic macroinvertebrates and algae have been

developed that provide consistent indications of biological condition across the diverse geography and climates in the state (*Fetscher et al., 2013*; *Mazor et al., 2016*; *Ode et al., 2016*). A physical habitat index has also been developed that provides complementary information supporting bioassessment data (*Rehn, Mazor & Ode, 2018*). Combined, these indices represent significant achievements in overcoming technical challenges for developing accurate and interpretable bioassessment products. However, these products are not used at a statewide scale to inform decisions and past efforts for stream management have only used a fraction of available products. A synthesis of condition assessments is needed to effectively implement bioassessment products in California and data must be presented in a context that is relevant to the needs of decision makers.

Recent regulatory initiatives in California have established a foundation for openness that could greatly improve the application of bioassessment products to support decision-making. In particular, these initiatives have set a precedent for openly sharing data collected with public funds. The Open and Transparent Water Data Act passed by the state legislature in 2016 requires water quality institutions to "create, operate, and maintain a statewide integrated water data platform that, among other things, would integrate existing water and ecological data information from multiple databases and provide data on completed water transfers and exchanges" (AB 1755, Dodd, 2015--2016). This legislation also calls for state agencies to "develop protocols for data sharing, documentation, quality control, public access, and promotion of open-source platforms and decision support tools related to water data". These aspirations were further supported by a resolution on July 10, 2018 that formally committed the State Water Resources Control Board to "provide broader access to data used to make local, regional, and statewide water management and regulatory decisions in California". These recent initiatives in California have similarly been observed at the national level. For example, the Data Coalition is an advocacy group that operates on behalf of the private and public sector for the publication of government data in a standardized and open format. The Internet of Water also operates at the national-level by focusing on strengthening connections between data producers and users through centralized data hubs and data standards.

Open science tools have recently been used in California to address bioassessment implementation challenges in developed landscapes. The Stream Classification and Priority Explorer, or SCAPE (*Beck, 2018a*; *Beck et al., 2019*), was developed using an open science framework to help identify reasonable management goals for wadeable streams using existing bioassessment and watershed data. The SCAPE tool represents both a modeling approach to help prioritize management goals (Fig. 3) and a set of open science products for direct application to environmental managers. The modeling component addresses a practical problem of achieving reference conditions in developed landscapes, where channel modification is common. Using the National Hydrography Dataset (NHD-Plus; *McKay et al., 2012*) and watershed predictors (StreamCat; *Hill et al., 2016*), the model classifies stream segments as biologically "constrained" or "unconstrained" by landscape alteration. This classification system can be used to set management priorities based on the constraint class. For example, a monitoring site with an observed biological index score that is above

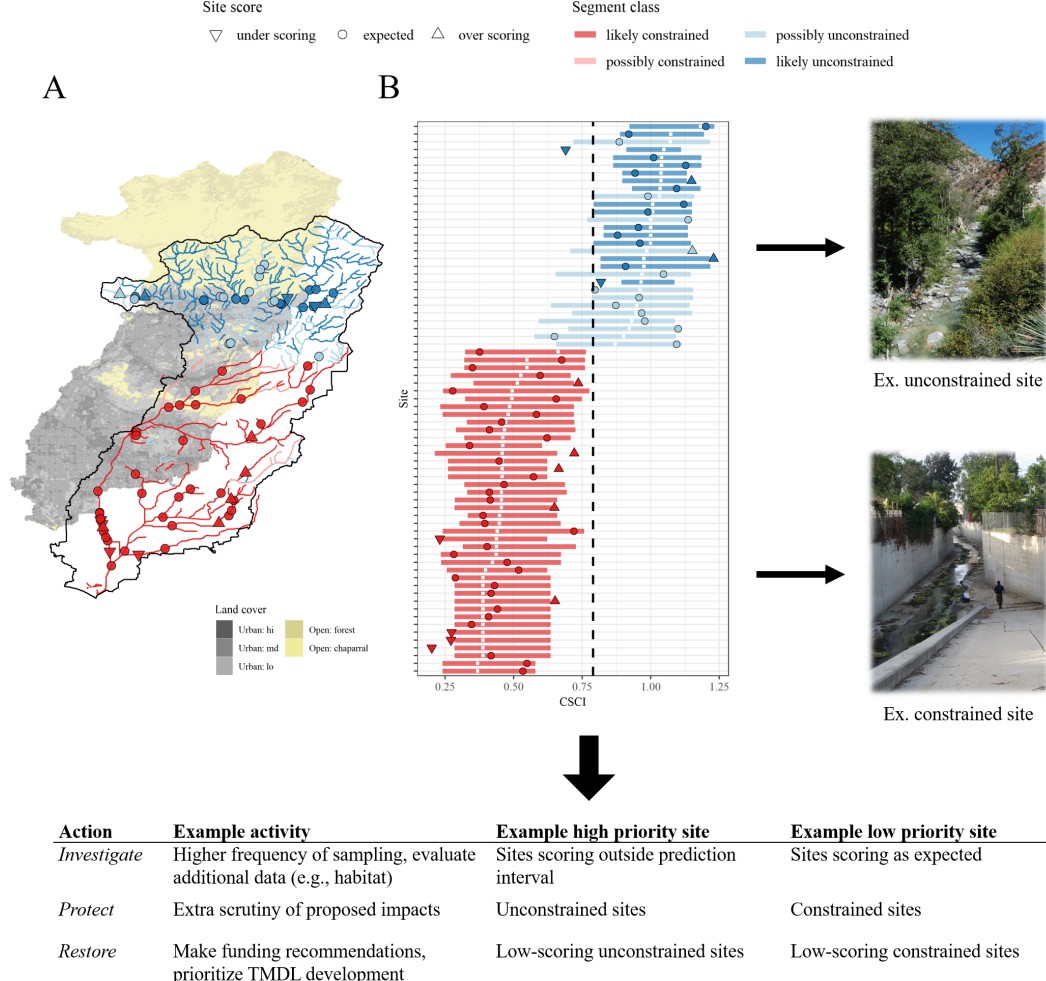

| Action | Example activity | Example high priority site | Example low priority site |
|---|---|---|---|
| *Investigate* | Higher frequency of sampling, evaluate additional data (e.g., habitat) | Sites scoring outside prediction interval | Sites scoring as expected |
| *Protect* | Extra scrutiny of proposed impacts | Unconstrained sites | Constrained sites |
| *Restore* | Make funding recommendations, prioritize TMDL development | Low-scoring unconstrained sites | Low-scoring constrained sites |

**Figure 3** **Schematic demonstrating how the Stream Classification and Community Explorer (SCAPE) can be used to identify potential management actions for stream sites.** Stream segment classifications are defined as biologically constrained or unconstrained based on landscape characteristics (A) and sites with bioassessment scores are evaluated relative to the classifications. Sites can be under-scoring, as expected, or over-scoring relative to the segment classification and expected range of scores (B). Unconstrained sites are those where present landscape conditions do not limit biological potential and constrained sites are those where landscape conditions limit biological potential. Management actions and priorities can be defined based on site scores relative to segment classifications. TMDL: Total Maximum Daily Load. Photo credit: Raphael Mazor.

a predicted range could be assigned a higher management priority relative to a site that is scoring within the range that is expected based on landscape development.

Open science tools were critically important for translating and delivering SCAPE products to decision-makers. Local stakeholder engagement to identify research goals guided the technical development process of SCAPE. All analyses, including model development and validation, were conducted using R. A version control system (Git) and online hosting (GitHub) also allowed full transparency of decisions that were made to create the SCAPE model. A permanent DOI was assigned through Zenodo to track

downloads and portability of source code (*Beck, 2018a*). Importantly, an online, interactive web page (https://sccwrp.shinyapps.io/SCAPE) greatly increased the impact and relevance of SCAPE by improving stakeholder understanding through direct interaction with key decision points that influenced model output. A manuscript describing the technical components of the model was created using `knitr` and `RMarkdown` (*Xie, 2015*; *Allaire et al., 2018*). This increased efficiency of the writing process also minimized the potential of introducing errors into tables or figures by eliminating the need to copy results between different writing platforms. Finally, a geospatial data file from the model was also made public on a federated data repository, which included metadata and plain language documentation to track provenance of the original information (*Beck, 2018b*).

## Limitations and opportunities

Although the case for open science in bioassessment is appealing, the widespread adoption of these principles in practice is inhibited by inertia of existing practices, disciplinary culture, and institutional barriers. Conventional and closed workflows used by many scientists are adopted and entrenched because of ease of use, precedence, and familiarity, yet they can be inefficient, inflexible, and difficult to communicate or replicate. Open science tools can improve analysis, documentation, and implementation through greater flexibility, but they expose research teams to entirely new concepts and skillsets in which they may never have been trained (e.g., *Idaszak et al., 2017*). Not only are the required skillsets demanding, but the open science toolbox continues to expand as new methods are developed and old methods become obsolete. This requires a research team to stay abreast of new technologies as they are developed and weigh the tradeoffs of adopting different workflows for different research tasks.

Advocates for open science are well aware of the technical challenges faced by individuals and research teams that have never been exposed to the core concepts. Most importantly, education and training (e.g., through The Carpentries) remain key components for developing skillsets among researchers where the focus is both on learning new skills for transferability and realizing their value for improving science as a whole (*Hampton et al., 2017*). A goal of many training curricula is to instill confidence in new users by developing comfort with new workflows, such as replacing a point-and-click style of analysis with one focused on using a command line through a computer terminal. Other approaches to demonstrate the value of new techniques use a side by side approach of closed vs open workflows to show the increased efficiency and power of the latter. Adoption becomes much more reasonable once users realize the value of investing in learning a new skill.

Advocates of open science also recognize the limitations of teaching in that not all audiences can be reached and not all materials are retained or even used after training. A strategy of empowering trainees to become trainers and teach others at their home institutions (e.g., train-the-trainer workshops and programs) enables open science to reach more individuals, and benefits science more broadly as they develop technical and communication skills, and build local communities. Those that also adopt new workflows through training can also direct their research products to facilitate collaboration with non-adopters rather than the latter synthesizing and analyzing their data in potentially

suboptimal ways (*Touchon & McCoy, 2016*). These "champions" can be a voice of encouragement for others by demonstrating how new tools can be introduced and learned over time through shared experiences (*Lowndes et al., 2017*). This also encourages the development of a community of practice that shares and learns together to navigate the collection of existing and developing open science tools (*Stevens et al., 2018*). Champions of open science should also be vocal proponents that spread awareness of the value of open science tools, particularly to those that make decisions on project resources. Department heads or administrative leaders may not be aware of the value of investing in open science, particularly if the consequences of not doing so are externalized in indirect costs that are not budgeted. A change in mindset may be needed where open science is viewed as a core tool that is critical to maintaining relevance of a research program in the future (*Hampton et al., 2017*).

Many scientists feel they cannot prioritize learning new skills given existing demands on their time, particularly if the benefits of these approaches, such as the value for the research team of sharing their data, are not apparent or immediate. Short-term funding and even political cycles can disincentivize scientists from spending time on anything but contractually obligated deliverables, which as noted above, may not effectively apply science in decision-making. This is an acute concern for early career scientists that have higher demands on establishing reputation and credentials, where investments in open science may be seen as detracting from progress (*Allen & Mehler, 2019*). As an alternative, a practical solution is to actively encourage the investment in open science within the research team or lab, as opposed to placing the burden on the individual as an isolated researcher (i.e., team science, *Cheruvelil & Soranno, 2018*). Laboratory or department heads should allow and encourage research staff to invest time in learning new skills and exploring new ideas, even if this does not immediately benefit the latest project. Over time, small investments in developing new skills will have long-term payoffs, particularly if a growing skillset among the research team encourages networking and peer instruction (*Lowndes et al., 2017*; *Allen & Mehler, 2019*). Developing an environment where open science tools are highly valued and encouraged may also increase job satisfaction and benefit recruitment and retention if researchers are allowed the space and time to develop skills beyond the current project.

The scientific culture within a discipline or institution may inhibit the adoption of open science methods. A common argument against open science is the protection of data that an individual research team may view as proprietary or sensitive. There are reasonable arguments to treat data as personal property, particularly if exceptional effort was spent to secure funding for a project and if the data were hard-earned or sensitive, e.g., detailed location data on endangered species or medical/socioeconomic data (*Zipper et al., 2019*). These issues are less of a concern for bioassessment where many datasets are collected by institutions that are publicly funded and data accessibility may be mandated by law. However, an open science process dictates that both interim and completed research products derived from public data should be available to the broader bioassessment community. This raises an additional concern that research teams using transparent workflows could expose themselves to increased criticism by their peers and the public

(*Lewandowsky & Bishop, 2016*; *Allen & Mehler, 2019*), particularly where the developed products can have important regulatory implications.

Feedback and criticism are fundamental and natural parts of the scientific process. Scientists receive feedback at many stages in the conventional scientific workflow (e.g., internal review, peer-review, presentations at conferences). Potentially new and challenging avenues for feedback are created in an open workflow. A concern is that openness can provide a platform for antagonistic or even hostile views, which could alter or degrade the scientific product (*Landman & Glantz, 2009*; *Lewandowsky & Bishop, 2016*). However, opportunities for addressing alternative viewpoints are critical to the open process of creating applied products, even if some voices are politically charged. This is especially true in bioassessment where finished products that could be adopted in regulation are often heavily scrutinized. It is in the interest of applied scientists to hear the concerns of all parties during the development phase. This is not to provide an avenue to erode the integrity or objectives of the science, but to enable full knowledge of the very real barriers to adoption that exist when science is applied in regulation. Openness that invites all voices to participate is a much more agreeable path to consensus than producing the science in isolation of those that it affects (*Pohjola & Tuomisto, 2011*). Ultimately, these products are developed to improve the environment as a public resource and the ideals promoted by an open science process directly align with these goals.

Institutional barriers can inhibit open science given the scale of change that must occur for adoption. Bureaucratic hurdles can disincentivize initiatives that promote change, particularly if that change originates from researchers not in administrative roles. Regulatory institutions may also prefer some level of opacity for how research products that influence policy are made available during development. The level of transparency advocated by open science could be viewed as opening the floodgates to increased legal scrutiny that can unintentionally hinder forward progress. Despite these reservations, many public institutions now advocate for increased openness because of the benefits that facilitate and engender public trust. Open data initiatives are now fairly common and represent a form of advocacy by public institutions for broader adoption of open science principles. Many national-level data products already exist that embrace openness to invest in the quality and availability of data (e.g., initiatives, US Geological Survey products through NWIS and BioData, US Environmental Protection Agency through STORET/WQX). Internationally, institutions in Europe and Canada that have projects supported by public funds are obligated to publish data and papers as open access (Horizon 2020, Tri-Agency Open Access Policy). Although past efforts and recent changes represent progress, many institutions have yet to strictly define open science and how it is applied internally and externally. As open science continues to build recognition, means of integrating toolsets that promote openness and transparency beyond publicly shared data will have to be adopted by regulatory and management institutions.

## CONCLUSIONS

The relevance of bioassessment applications can be improved with open science by using reproducible, transparent, and effective tools that bridge the gap between research and

management. Many open science tools can improve communication between researchers and managers to expose all aspects of the research process and facilitate implementation to support policy, regulation, or monitoring efforts. Communication ensures that the developed product is created through an exchange of ideas to balance the potentially competing needs of different sectors and institutions. The documentation and archiving of data used to create a bioassessment product also ensures that other researchers can discover and build on past efforts, rather than constantly rebuilding the wheel. Incremental improvements of existing products can reduce the proliferation of site- and taxon-specific methods with limited regional applications by exploring new ways to integrate biological indicators across space and time.

Efforts to formally recognize and integrate open science in bioassessment are needed now more than ever. The transition of bioassessment from taxonomic-based indices to molecular approaches presents novel challenges that will only increase in severity as researchers continue to refine methods for molecular applications (*Baird & Hajibabaei, 2012*). Although molecular-based indices share similar assessment objectives as conventional indices, the data requirements and taxonomic resolution are substantially more complex. Bioassessment researchers developing molecular methods are and will continue to be inundated with data from high-throughput DNA sequencers. Systematic approaches to document, catalog, and share this information will be required to advance and standardize the science. Molecular approaches are also dependent on existing reference libraries for matching DNA samples for taxonomic identification. The integrity of reference libraries depends greatly on the quality of metadata and documentation for contributed samples. Open science principles should be leveraged in this emerging arena to ensure that new bioassessment methods continue to have relevance for determining the condition of aquatic resources.

## ACKNOWLEDGEMENTS

The authors acknowledge technical support through an advisory committee from the San Gabriel River Regional Monitoring Program and the Sanitation Districts of Los Angeles County. We thank Eric Stein for reviewing an earlier draft of the manuscript. We thank Mike McManus for assistance with the conceptual diagrams. We thank two anonymous reviewers for providing excellent feedback that improved the manuscript.

### Funding

The authors received no funding for this work.

### Competing Interests

The authors declare there are no competing interests. MW Beck is employed by the Tampa Bay Estuary Program, but completed the manuscript as an employee of the Southern California Coastal Water Research Project (SCCWRP). Other non-academic affiliations

include RD Mazor, S Theroux, DJ Gillett employed by SCCWRP and G Gearheart employed by the California State Water Resources Control Board.

## Author Contributions

- Marcus W. Beck and Casey O'Hara conceived and designed the experiments, analyzed the data, prepared figures and/or tables, authored or reviewed drafts of the paper, and approved the final draft.
- Julia S. Stewart Lowndes conceived and designed the experiments, prepared figures and/or tables, and approved the final draft.
- Raphael Mazor conceived and designed the experiments, analyzed the data, prepared figures and/or tables, and approved the final draft.
- Susanna Theroux and David Gillett conceived and designed the experiments, authored or reviewed drafts of the paper, and approved the final draft.
- Belize Lane performed the experiments, authored or reviewed drafts of the paper, and approved the final draft.
- Gregory Gearheart performed the experiments, analyzed the data, authored or reviewed drafts of the paper, and approved the final draft.

## Data Availability

All source materials for reproducing the paper are available in GitHub: https://github.com/fawda123/bioassess_opensci.

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
