# Peer review of "The importance of open science for biological assessment of aquatic environments"

_PeerJ, doi:10.7717/peerj.9539_

## Round 0.1 · original submission · Major Revisions

Dear Marcus and co-authors,

I have received three valuable assessments or your original review paper. All reviewers have recognised that the manuscript is of good quality, informative, timely, and within the scope of the journal.

However, a number of important considerations have been raised and need to be fully addressed in the next version. In particular, narrowing down the title of the review to 'aquatic biological assessment' (reviewers 1 and 2), re-tailoring the introduction and discussion sections to a more prominent mention of the failures in reliability that brought about the open science movement (reviewer 1) while also expending the background beyond the USA context (reviewer 2), and perhaps providing more emphasis on what needs to be done for environmental managers to effectively train and apply ecogenomics and R-based bio-assessments (reviewer 3).

Overall, the reviewers have provided you with excellent suggestions on how to improve this study, and I will be looking forward receiving your revised manuscript along with a point-by-point response to their comments.

With warm regards,
Xavier

Reviewer 1 ·

Basic reporting

This article presents a review of open science practices in ecological science. I found it to be well written and informative. I did however find the approach taken to be quite limited to ecological aquatic biological assessment. I understand within the field biological assessment is the correct term but given the scope of the journal, biological assessment could be expected to include other areas of biology such as medicine. Perhaps clear statement of ecological angle in the title would resolve this issue. I also thought the introduction would benefit from a more prominent mention of the failures in reliability that brought about the open science movement. In its current form, it seems to advocate standardisation as much as open practices (although I see the two are related). I also found the description of the principals of open science to be slightly narrower than I had expected (see below). I am not aware of a similar review having been recently published.

Experimental design

As stated above I found the coverage to be somewhat limited to ecological methodologies, but this is probably acceptable if the scope is clearer at the outset. More importantly, a major aspect of open science practice more broadly is the adoption of study pre-registration and registered reports. I thought the absence of a discussion of the impact of these practices was an omission. I would also suggest that were the authors to include a discussion of these practices it would be advisable to enlist the aid of someone with first-hand experience of implementing them. Adopting these practices is a major cultural shift in open science, which I don’t think can easily be left out (including them would certainly alter figures 1 and 2). Perhaps the distinction between confirmatory and exploratory research is not as relevant here as in other disciplines (e.g. neuroscience) and environmental science does not rely on hypothesis testing as much, in which case might that be noted? I also thought broader discussion of open access platforms and repositories (such as the osf.io) would benefit the reader.

Validity of the findings

The conclusions appear to be well supported and clearly made. My concerns here are centred upon the apparent omission of important aspects to open science practices, mentioned above.

Additional comments

I found this to be a manuscript of good quality. I did, however, feel a number of important open science practices, partially pre-registration, where not discussed and perhaps should have been. Perhaps the authors could consider narrowing the scope of the article and discussing open data and materials practices, rather than open science?

Reviewer 2 ·

Basic reporting

The English is clear, unambiguous, and professional.
Although the Introduction and background show the context, there are some issues which need to be taken into account, including the absence of some relevant literature. Some comments to the Introduction are shown below.
- Line 1: I would add ‘aquatic’ in the title, before ‘biological assessment’, since the authors focus mainly in that and this should be highlighted since the very beginning
- Line 24: ‘Democratize science’, probably this is an opinion of the authors, and my response could be considered also as an opinion. As such, you can take it into account or not. I don’t like this expression, since it seems that the general scientific process is not democratic. However, in my opinion, is one of the most scrutinized human activities, since the moment in which scientists apply to public calls for projects and then publish the results either in open access or non-open access journals, since they are checked by editors and reviewers, trying to ensure replicability of the publications. Are open access principles more ‘democratic’? I disagree: they allow more interaction, more replicability, more transparent, more responses to new questions,… but I cannot see that as more ‘democratic’.
- Lines 42-46: it is good to have some examples of the legislation driving monitoring and assessment in these three areas. However, there are many other countries/regions (e.g. China, South Africa, Australia, New Zealand, etc.), as well as other legislation (e.g. Oceans Act in several countries, Marine Strategy Framework Directive, international conventions, etc.), which deserve also to be cited. As this could be too long, I suggest you adding a citation here. To my best knowledge, although it has more than 10 years, the most complete review in this specific topic is this paper, collating the legislation around the world in monitoring and assessment: Borja, A., S. B. Bricker, D. M. Dauer, N. T. Demetriades, J. G. Ferreira, A. T. Forbes, P. Hutchings, X. Jia, R. Kenchington, J. C. Marques, C. Zhu, 2008. Overview of integrative tools and methods in assessing ecological integrity in estuarine and coastal systems worldwide. Marine Pollution Bulletin, 56: 1519-1537.
- Lines 48-49: regarding marine waters, one of the co-authors of this manuscript has also a paper that can be cited here, since it constitutes a good review of the current methods availability: Borja, A., M. Elliott, J. H. Andersen, T. Berg, J. Carstensen, B. S. Halpern, A.-S. Heiskanen, S. Korpinen, J. S. S. Lowndes, G. Martin, N. Rodriguez-Ezpeleta, 2016. Overview of integrative assessment of marine systems: the Ecosystem Approach in practice. Frontiers in Marine Science, 3: doi: 10.3389/fmars.2016.00020.
- Lines 72-85: regarding the use of open science in bioassessment, in relation to stakeholders, to my knowledge, this paper below is the single one dealing with that. The second one is dealing with the transparency issues. Both could be useful in this context.
o Borja, A., J. M. Garmendia, I. Menchaca, A. Uriarte, Y. Sagarmínaga, 2019. Yes, We Can! Large-Scale Integrative Assessment of European Regional Seas, Using Open Access Databases. Frontiers in Marine Science, 6: 10.3389/fmars.2019.00019.
o Essl, F., F. Courchamp, S. Dullinger, J. M. Jeschke, S. Schindler, 2020. Make Open Access Publishing Fair and Transparent! BioScience, 70: 201-204.

The structure is clear, and of broad and cross-disciplinary interest.
I don’t know any other recent review. However, one of the co-authors of this paper, is the first author or another paper on the topic (Lowndes et al., 2017), included in the references list. Probably, there are other papers around the topic, such as that abovementioned of Borja et al. (2019) and few others (e.g. van Oudenhoven, A. P. E., M. Schröter, R. de Groot, 2016. Open access to science on ecosystem services and biodiversity. International Journal of Biodiversity Science, Ecosystem Services & Management: 1-3).

Experimental design

The article content is within the Aims and Scope of the journal. Regarding the methods described, I think that they need more explanations:
- Line 87: for the whole section, I cannot see later in the manuscript the results obtained from that search in Google Scholar. Why including this section, if there is no a section on the results? (maybe I have missed something).
- Lines 104-107: a justification of the selection of these terms is necessary. There are many other possible that could expand the number of potential papers. Some few examples: ‘open databases’, ‘assessment’, ‘ecological status’, ‘health status assessment’, etc.
- Lines 108-109: I agree that they are scarce, but an example that the terms used are insufficient is the fact that the first paper abovementioned was not found.
Although the review is organized logically, into coherent paragraphs/subsections, I cannot see why those subsections (what is the logic behind that organization, or what is the source to organize it in that way).

Validity of the findings

Some comments about the main text body.
- Line 134: Figure 1: You should add the word ‘open’ in ‘publish data’ and ‘publish code’. In addition, in this conceptual workflow I think that something is wrong: after the publication in open access, you link the arrow with ‘conceptualize project’. However, if you are discussing about bioassessment, monitoring and taking decisions to improve aquatic systems, I think that, after publication, at least two boxes should be added: one on taking management measures, and another on recovery of the systems monitoring, which can lead to new ‘conceptualization of projects’. Otherwise, this figure is not useful, and endogamic for the research community.
- Lines 167-170 and 188-196: an example of this are the metabarcoded databases of species (e.g. BOLD, GenBank,…).
- Lines 253-255: there are also many taxonomical resources that can be mentioned e.g. WoRMs
- Lines 400-417: you can mention also that in Europe any project supported by public funds must provide open access data and publish all papers in open access: https://ec.europa.eu/research/participants/docs/h2020-funding-guide/cross-cutting-issues/open-access-dissemination_en.htm. The same in Canada: https://www.ic.gc.ca/eic/site/063.nsf/eng/h_F6765465.html, and probably in many other countries/regions. Avoid being so local.
- Line 424: spell out, in the legend of Figure 3, the abbreviation: TDML
The conclusions are coherent

Additional comments

The manuscript has some interest, but I have identified some weaknesses, regarding the lack of connection between the methodology presented and the absence of results associated to the search in Google Scholar. In addition, some references that I have included above could be of interest. The authors should present the applicability more global, not as local as California (although it can serve as example, I have given more examples in other areas)

Reviewer 3 ·

Basic reporting

no comment

Experimental design

no comment

Validity of the findings

no comment

Additional comments

This is an important and innovative paper that highlight the issue of the accessibility of scientific results and data to facilitate the bioassessment of environmental impacts. The paper is very timely as the open science is becoming a standard for most of biological studies and it’s important that this practice is also adopted by ecological studies of practical importance for environmental managers.

The review is very well written, and I fully recommend its publication. I would suggest however few points to be considered in the revision of this paper.

The first point concerns the new ecogenomic tools that are currently transforming the field of biomonitoring and bioassessment. I was quite surprised to find no reference to these tools despite the fact that their impact on so-called “democratization” of bioassessment practices is well documented (e.g. Deiner et al. 2017, Pawlowski et al. 2018, Hering et al. 2019). In particular, it is worth to be mentioned that the access to genomic or metagenomic data can be much easier and more open compared to the classical morphotaxonomic data, which are more depending on personal expertise and more difficult to compare with others. The authors could address this point when they are discussing the issue of species identification (subchapter Curating bioassessment data).

The second point, somehow related to the first, concerns a certain conservationism of bioassessment practitioners regarding scientific progress. As discussed by Guareschi et al. 2019, this is not only about new tools but also about ecological changes, such as the presence of alien species, or changes in taxonomic identification. Of course, open science can help pointing out these changes and interpreting them correctly. However, even the most accessible research will do nothing if the environmental managers are not particularly interested in getting updated about the new research advances and implementing them in their practice.

My final critical remark concerns the scope of the review and its emphasis on open source software applications. The authors provide a very exhaustive description of using R for bioassessment, but they go into a lot of details that somehow contrast with a general character of the review. They assume most of environmental managers know how to use R, which I doubt is true. In fact, the authors seem to agree with me, when they point out that the education and training are the main limitations of the access to the open science tools and data. It’s probably the main message of this review paper and I would rather focus on it than on describing the various applications of tools that most probably remain inaccessible for most of practitioners.

References:
Hering et al. 2019
https://www.sciencedirect.com/science/article/pii/S0043135418301830?via%3Dihub
Guareschi et al. 2019
https://www.sciencedirect.com/science/article/pii/S0048969719300087?via%3Dihub
Deiner et al. 2017
https://onlinelibrary.wiley.com/doi/full/10.1111/mec.14350

---

## Round 0.2 · Minor Revisions

I just wanted to first let you know that I am taking over handling of this manuscript from Prof. Pochon. Having read the revised manuscript and the response from the reviewers, I am satisfied that you have addressed all of the concerns and that the manuscript is now acceptable for publication.

I return it to you as minor revisions only to allow you the opportunity to incorporate the constructive feedback of Reviewer #1 from this revision if you so desire. I do not expect that it will have any additional review, and will be happy to move it forward into production after you have the opportunity to consider these additional comments and make any final changes that you wish to the final manuscript.

Reviewer 1 ·

Basic reporting

The authors have addressed the concerns raised well. I think the article is a timely, well written, and informative addition to the literature that I would like to recommend this for publication.
I have a few mirror comments with regards some of the new sections that the authors may wish to consider, if they are given the opportunity, but I do not think these changes are completely necessary.
In the new section describing pre-registration (from line 264), it might be worth mentioning that some journals offer grantee of publication (if protocols are stuck to) irrespective of null findings. This is what prevents publication bias, not pre-registration per se. Most journals differentiate between pre-registration (which guards against other questionable research practices such as p-hacking and HARKing) and Registered Reports which are a type of pre-registration but with the additional guarantee and peer reviewer from a journal prior to data collection. If the authors wishes to include this differentiation I think it might help the readers understanding. However, I appreciate that these terms are often used interchangeably. I also appreciate that this does not site perfectly with the scheme set out in figure 1, but as Registered Reports are becoming more broadly accepted, with over 240 participating journals, including many specialising in ecology (see https://www.cos.io/our-services/registered-reports?_ga=2.14444030.83877480.1591349264-1270378436.1498895408), I would recommend mention of them.
Also, the authors differentiate between exploratory and applied research implying that pre-registration could be useful for declaring research to be exploratory (lines 277-280). I found this confusing, although again this is likely to be subject specific. Commonly, amongst the open science community, the relevant distinction is between exploratory (i.e. not pre-registered, where data is explored) and confirmatory (pre-registered) analyses.

Experimental design

Fine - as before.

Validity of the findings

Fine - as before.

Additional comments

Fine - as before. See basic reporting comment.

Reviewer 2 ·

Basic reporting

I have carefully read the responses to my comments and those from the other reviewers, as well as the new version of the manuscript and the reviewers have addressed satisfactorily them. The new version of the manuscript has improved and now can be published.

Experimental design

the comments have been addressed adequately

Validity of the findings

the comments have been addressed adequately

---

## Round 0.3 · accepted · Accept

Thank you for making those final updates to the manuscript. I am happy to now move your revised manuscript forward into production.
Congratulations, and thank you for selecting PeerJ to publish your work!